

# A novel methodology for optimal land allocation for agricultural crops using Social Spider Algorithm

N Thilagavathi[*] and T Amudha[*]

Department of Computer Applications, Bharathiar University, Coimbatore, Tamil Nadu, India
[*] These authors contributed equally to this work.

## ABSTRACT

In the current agricultural scenario, availability of suitable land for cultivation is less and profitable allocation of the land for cultivating crops seems to be a cumbersome task. Crop planning optimization is a major research field in agriculture, in which land optimization is a significant challenge, which falls under the category of combinatorial optimization problems. The main objective of the present research is to maximize the net income from agriculture through optimal land allocation. Bio-inspired algorithms are quite popular in solving combinatorial optimization problems. Social Spider Algorithm (SSA), a new bio-inspired algorithm, is used to solve land optimization problem in this research based on the simulation of cooperative behaviour of social spiders. The agricultural area chosen for case study is the Coimbatore region, located in Tamilnadu state, India and the relevant data for the crops are collected from Tamilnadu Agricultural University Coimbatore, India. The optimal planting area, crop productivity for various land holdings and the water requirements are computed by SSA and the results have shown better directions for agricultural planning to improve the profit with constrained land area and water limitations.

# INTRODUCTION

Agriculture is the main source of revenue in India. As per the World Bank data, around 60.6% of India's land area is used for agriculture (https://data.worldbank.org/), but the importance of agriculture is not realized by most of the population. Moreover, very few people have opted for agriculture as their occupation and most of the agricultural lands are converted into residential lands or utilized for industrial purpose (*Dussal, 2012*). Hence, the agricultural sector faces many challenges to meet the global food requirements. Crop planning optimization is one among the major challenges, which needs to be addressed. The current agricultural system practices mono-cropping with more fertilizers, hybrid seeds and pesticides. In agriculture, optimization can be performed on irrigation, crop planning, land, soil, labour, transportation, climate, weed, fertilizer and pesticides.

Corresponding authors
N Thilagavathi,
thilagamca86@gmail.com
T Amudha,
amudhaswamynathan@buc.edu.in

Various objectives and constraints can be considered for every type of optimization (*Adekanmbi, 2014*).

Present research focuses on agricultural land optimization which is the need of the hour in agricultural sector. Land optimization faces challenges such as allocating land among multiple crops based on the season (Kharif (June–October) and Rabi (October–April)), soil type, irrigation water, crop growing duration, climate, land availability and many other factors. Crops can also be categorised as agricultural crops, plantation crops, horticulture crops, forage crops and manure crops. Agricultural crops are considered for optimization in this work (*Raghava Rani & Dr Tirupathi Rao, 2012*).

Currently, farmers are cultivating crops, applying fertilizer and irrigating the land randomly based on their assumptions. The available land is also not properly utilized for cultivating suitable crops. Most of the Indian farmers are following the traditional methods for crop planning. Farmers face heavy economic loss due to the lack of awareness in utilizing the available land and water resources. To avoid all such issues, flexible and suitable cropping patterns and irrigation methods can be taken up by the farmers, which will have a positive impact towards their economy. This research work is focussed on suggesting the suitable crops for cultivation, based on the on the available land area and water resources. It is also focussed on improving the profit to the farmers by optimally allocating the available land without leaving much of unused land, along with minimum water requirements (*Thilagavathi, Amudha & Sivakumar, 2017*).

The main objective of this research is to maximize the total profit and minimize the water demand by using Social Spider Algorithm. The problem should meet some constraints such as total cropping area and minimum land allocation for crops (*Wankhade & Lunge, 2012*).

## METHODS

### Study area

The study area considered for the current research is Coimbatore district, which is located in the western agro climatic zone. It lies between $11°01'06.00''$N latitude and $76°58'2900''$E longitude [TN18-COIMBATORE 31.03.2011]. The district is divided into 9 taluks, 19 blocks and 481 villages. The main rivers flowing through Coimbatore district are Bhavani, Noyyal, Amaravathi, Aliyar, Nirar, Solaar and Uppar Thirumurthi. The total area of Coimbatore district is 7,470 square kilometres, among which 3,319 square kilometres are used as cultivable areas. The major crops are millets, jowar, paddy, cowpea, bengal gram, horse gram, green gram, coconut, groundnut, sugarcane, and cotton and the main soil types are red calcareous soil, red non-calcareous soil, black soil, alluvial and colluvial soil.

## MATERIAL AND METHODS

The primary objective of the present research is to maximize the total profit (TP) from agriculture through optimal allocation of the available agricultural land for planting suitable crops. This research identifies the feasible combinations of crops that could be grown in

the specified land area and assigns the available land optimally for these crops, thereby improving the production.

## Objective 1: Profit maximization

The objective function to maximize the total profit is given in Eq. (1).

$$\text{Max TP} = \sum_{i=1}^{n} A_i [I_i - Exp_i] \tag{1}$$

where TP is the total profit that could be obtained from optimal allocation of land for crops, $n$ is the total number of crops, $i$ is the crop which varies from 1 to $n$, $A_i$ is the optimal area allocated to $i$th crop (square metres), $I_i$ is the income from $i$th crop, $Exp_i$ stands for expenses incurred for crop $i$.

$I_i$ is calculated by using the price and production of the selected crop i, $price_i$ and $productivity_i$, as given in Eq. (2). The production value of the crop, $pro_i$ is calculated as given in Eq. (3), where the productivity values of the crops, $productivity_i$ are collected from the website [www.coimbatore.nic.in/pdf/SHB003.pdf].

$$I_i = Price_i \times Productivity_i \tag{2}$$

$Exp_i$ includes expense on seeds, expense on manure, expense on fertilize, expense on irrigation and expense on labour as given in Eq. (3).

$$Exp_i = CS_i + CM_i + CF_i + CI_i + CHL_i \tag{3}$$

where $CS_i$ is the expense on seeds for $i$th crop, $CM_i$ is the expense on manure for $i$th crop, $CF_i$ is the expense on fertilizer for $i$th crop, $CI_i$ is the expense on irrigation water for $i$th crop and $CHL_i$ is the expense on labour and machinery for $i$th crop.

To achieve maximum profit, improved production from the crops and optimal allocation of land for the crops are highly essential. Crop production is obtained from Eq. (4).

$$\text{Max Production}_i = A_i \times productivity_i. \tag{4}$$

## Objective 2: Water requirement minimization

The secondary objective of this research is to minimize the usage of water to be used for irrigating the allocated crops due to the insufficient availability of water in the study area.

$$\text{Minimize TWR} = \sum_{i=1}^{n} (CWR_i \times A_i) \tag{5}$$

where TWR is the total water requirement (cubic metres) and $CWR_i$ is the crop water requirements for crop $i$. The constraint used is the total cropping area, which should be in the range of 2 to 4 hectares as given in Eq. (6).

## Constraint 1: Total cropping area

$$max.area \geq (TCA = \sum_{i=1}^{n} A_i) \geq min.area \tag{6}$$

where TCA is the total cropping area, max. area is the maximum available cropping area and min. area is minimum available cropping area. Two categories of land holdings taken in this research work are 20,000 $m^2$–40,000 $m^2$, which is called small-medium land holding category and 40,000 $m^2$–1,00,000 $m^2$, which is called medium land holding category.

## Constraint 2: Cropping area for each crop

Cropping area for each crop is constrained as given in Eqs. (7) and (8). Eq. (7) is used for small-medium land holding category and Eq. (8) is used for medium land holding category.

$$A_i \geq 2,000 \text{ m}^2 \tag{7}$$

$$A_i \geq 4,000 \text{ m}^2. \tag{8}$$

## Social Spider Algorithm (SSA) for crop planning optimization

SSA is a population based algorithm proposed by *Cuevas et al. (2013)* to solve global optimization problems. It is based on the reproduction and cooperative behaviour of social spiders. SSA possesses certain unique characteristics, which makes it quite different and competent than many other existing bio-inspired algorithms. The algorithm employs two types of spiders as search agents, male spiders and female spiders. Each individual performs different types of operations based on their type that simulate the social behavior of the spiders within the colony. Three types of vibration operators used in SSA allow improved particle distribution in the search space, thereby enhancing the algorithm's ability to find the global optima. Based on the vibration values, the male cooperative behavior and female cooperative behavior are determined in SSA, which in turn applies different mechanisms for exploration and exploitation during the evolution process. Another unique quality of this algorithm is the choice of new spiders for the next generation. While most of the algorithms use only the fitness of the offspring to calculate their chance to be moved to the next generation, SSA calculates and applies an influence probability in addition to the fitness, in order to assess the suitability of the individual to become a member of the new generation.

The major components of social spider colony are social members (spiders) and communal web (spider web). The communal web is considered as search Space (S). The position assigned to each spider on the web is based on its weight and fitness value (*Cuevas et al., 2013*). The total colony members are divided into two categories: 70% of female spiders and 30% of male spiders. The male spiders are subdivided into two classes; dominant and non-dominant males based on the fitness of the spiders. The mating operation allows the information exchange among colony members, performed by dominant male and females. The spiders generate vibrations for mating operation based on the other spider's weight and distance between each other. The dominant male spiders mate with one or more female spiders to produce offspring (new spider). The weight of the spider is considered as the fitness of the solution. The population contains the female ($f_i$) and male ($m_i$) spiders. The number of female spiders is randomly selected within the range of 65%–90% and the number of male spiders is calculated from female spiders based on Eqs. (9) and (10).

$$N_{fs} = \text{floor}[(0.9 - \text{rand}(0, 1) * 0.25) * P] \tag{9}$$

$$N_{ms} = P - N_{fs} \tag{10}$$

where $N_{fs}$ means number of female spiders, $N_{ms}$ represents the number of male spiders, $P$ stands for entire population and floor represents the real number to an integer number. The weight ($w_i$) of each spider is calculated based on (11). The fitness value obtained by the spider at position $s_i$ and the values $b_s$ and $w_s$ are calculated based on Eqs. (12) and (13) for maximization problems.

$$W_i = \frac{J(s_i) - w_s}{b_s - w_s}. \tag{11}$$

The $b_s$ is maximum fitness value and $w_s$ is minimum fitness value obtained from the fitness values of all the spiders.

$$b_s = \max_{k \in 1,2,\ldots,N} J(s_k) \tag{12}$$

$$w_s = \min_{k \in 1,2,\ldots,N} J(s_k). \tag{13}$$

Similarly, vibration of each spider is calculated based on the weight ($w_i$) and distance ($d_{i,j}$) between the individual $i$ and member $j$ as given in Eq. (14).

$$Vb_{i,j} = w_j . e^{-d_{i,j}^2} \tag{14}$$

where $d_{i,j}$ is the Euclidian distance between the spiders $i$ and $j$, which is calculated by using Eq. (15).

$$d_{i,j} = \sqrt{(x_i - x_j)^2 + (y_i - y_j)^2}. \tag{15}$$

Vibrations are categorized into three types based on the relationship between the pair of individuals. These three types of vibrations are calculated to perform the cooperative behaviours of male and female spiders.

- *Vibrations $Vc_i$*—the information transmitted from individual $i$ to member $c(s_c)$, which is the nearest member to individual $i$ based on Eq. (16).

$$Vc_i = w_c . e^{-d_{i,c}^2} \tag{16}$$

- *Vibrations $Vb_i$*—the information is transmitted between the individual $i$ and the best member $b(s_b)$ by using Eq. (17).

$$Vb_i = w_b . e^{-d_{i,b}^2} \tag{17}$$

- *Vibrations $Vf_i$*—the information is transmitted between the individual $i$ and the nearest female $s(s_f)$ as given in Eq. (18).

$$Vf_i = w_f . e^{-d_{i,f}^2} \tag{18}$$

The vibrations $Vb_i$ and $Vc_i$ are used to calculate the female cooperative behaviours while $Vf_i$ is used to calculate the male cooperative behaviours. Strong vibrations are produced by

the nearest spiders or big spiders.

$$f_i^{k+1} = \begin{cases} \{f_i^k + \alpha.vc_i.\left(s_c - f_i^k\right) + \beta.vb_i.\left(s_b - f_i^k\right) + \delta.\left(rand - \dfrac{1}{2}\right) \\ \text{with probability PF} \\ \{f_i^k - \alpha.vc_i.\left(s_c - f_i^k\right) - \beta.vb_i.\left(s_b - f_i^k\right) + \delta.\left(rand - \dfrac{1}{2}\right) \\ \text{with probability } 1 - PF \end{cases} \quad (19)$$

$$m_i^{k+1} = \begin{cases} m_i^k + \alpha.vf_i.\left(s_f - m_i^k\right) + \delta.\left(rand - \dfrac{1}{2}\right) \text{ if } w_{N_f+i} > w_{N_f+m} \\ m_i^k + \alpha.\left(\dfrac{\sum_{h=1}^{N_m} m_h^k.w_{N_f+h}}{\sum_{h=1}^{N_m} w_{N_f+h}} - m_i^k\right) \text{ if } w_{N_f+i} \leq w_{N_f+m} \end{cases} \quad (20)$$

where, $\alpha, \beta, \delta$ represents random numbers between $[0, 1]$, $k$ stands for iteration number and $s_c$ & $s_b$ means the nearest member to $i$ that holds the highest weight and the best individual of the entire population.

Equation (19) is used to calculate the female cooperative behaviours while Eq. (20) is utilized to calculate the male cooperative behaviours based on the threshold value. The spider movement, i.e., attraction or repulsion, is based on the random phenomena, which is represented as $r_m$, and it is generated within the range of $[0, 1]$. The $r_m$ value is compared with the threshold value, and if the value is less than $r_m$, then the attraction movement will be produced, else the repulsion movement will be produced.

$$r = \frac{\sum_{j=1}^{n}\left(P_j^{high} - P_j^{low}\right)}{2.n} \quad (21)$$

where $r$ is a range to select the female spiders for mating operation, which is calculated by using Eq. (21). The dominant male spiders are arranged in set $m_g$ andthe selected female spiders are arranged in set $E^g$. The sets $m_g$ and $E^g$ ($m_g \cup E^g$) are combined to get the reproduction. If the set $E^g$ is empty, then there will not be any mating operation.

$$Ps_i = \frac{w_i}{\sum_{j \in T^k} w_j}. \quad (22)$$

The influence probability Psi is calculated by using the weight of each spider as given in Eq. (22). After calculating the probability, the roulette wheel method is applied to select the $s_{new}$. The weight of the new spider $s_{new}$ is calculated and compared with the weight of population member. If the weight of $s_{new}$ isbetter than the weight of any member in the population, then the member will be replaced by $s_{new}$. Figure 1 depicts the search space of spiders denoted by S.

The present research applies SSA for crop planning optimization. The number of crops is considered as the population size i.e., the number of spiders. Ten agricultural crops are selected for implementation in this work. Total population is split into males and females by using Eqs. (9) and (10). The value of $p_j^{low}$ and $p_j^{high}$ are equated for the minimum area and maximum area of cropping land considered. The social spiders are responsible for allocating optimal land for each crop and based on the land allocation, the total profit from agriculture and TWR (objective functions) are calculated. Data for the cost of seed, cost

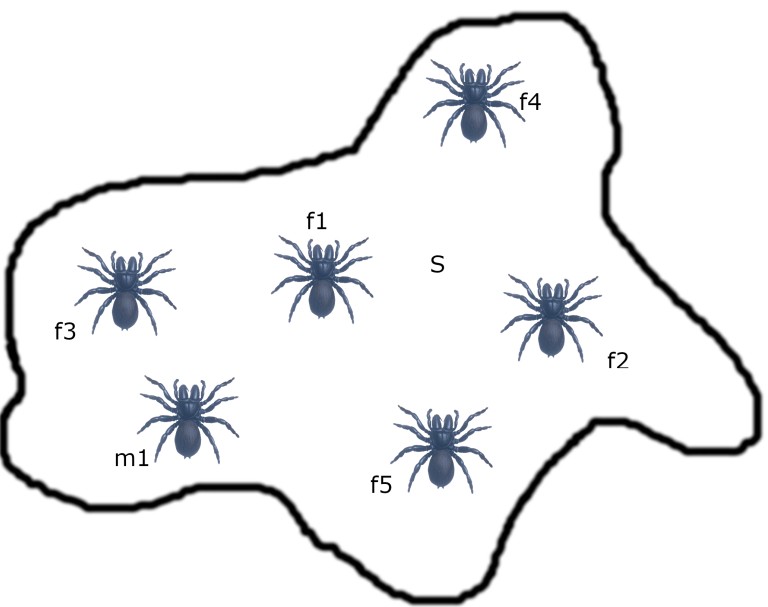

**Figure 1 Search space (S) of spiders.**

of irrigation, cost of manure, cost of fertilizer, cost of labour and machinery are obtained from the Centre for Agriculture and Rural Development Studies (CARDS), Tamilnadu Agricultural University, Coimbatore.

The weight and fitness of each spider are calculated from the objective function given in Eq. (11). The best and worst values of objective function are used for the weight calculation of each spider. Distance between each spider is calculated by utilizing Eq. (15). The values taken for the distance calculation are the weight and the total land allocation done by each spider. Similarly, vibration is calculated based on the distance between spiders. Three types of spiders chosen for the calculation of vibration are the spider with minimum distance ($Vbc_i$), spider with highest weight ($Vbb_i$) and nearest female spider ($Vbf_i$). Based on the vibrations, female cooperative operators and male cooperative operators are applied. The median male spider is found based on the weight to calculate the dominant and non-dominant male spiders. The spiders, whose weights are above the median male spider, are called the dominant males and those below the median are non-dominant males.

Cooperative operations are used to select the best female spiders and dominant male spiders to mating operation. The range value of dominant male spider is calculated to choose female spiders among female co-operative spiders to mating operation. Mating operation forms a new spider (brood) denoted as $s_{new}$. Each member of the new spider population suggests an optimal land allocation for multiple crops. Based on the suggestions, the objective function values are derived and the corresponding fitness and weight of the population members are calculated. The weight of the new members is compared with the weight of all the members in the population. If the new member is better than the worst member in the population, then the worst member will be replaced by the new member. This process will be continued till the optimum solution is obtained which

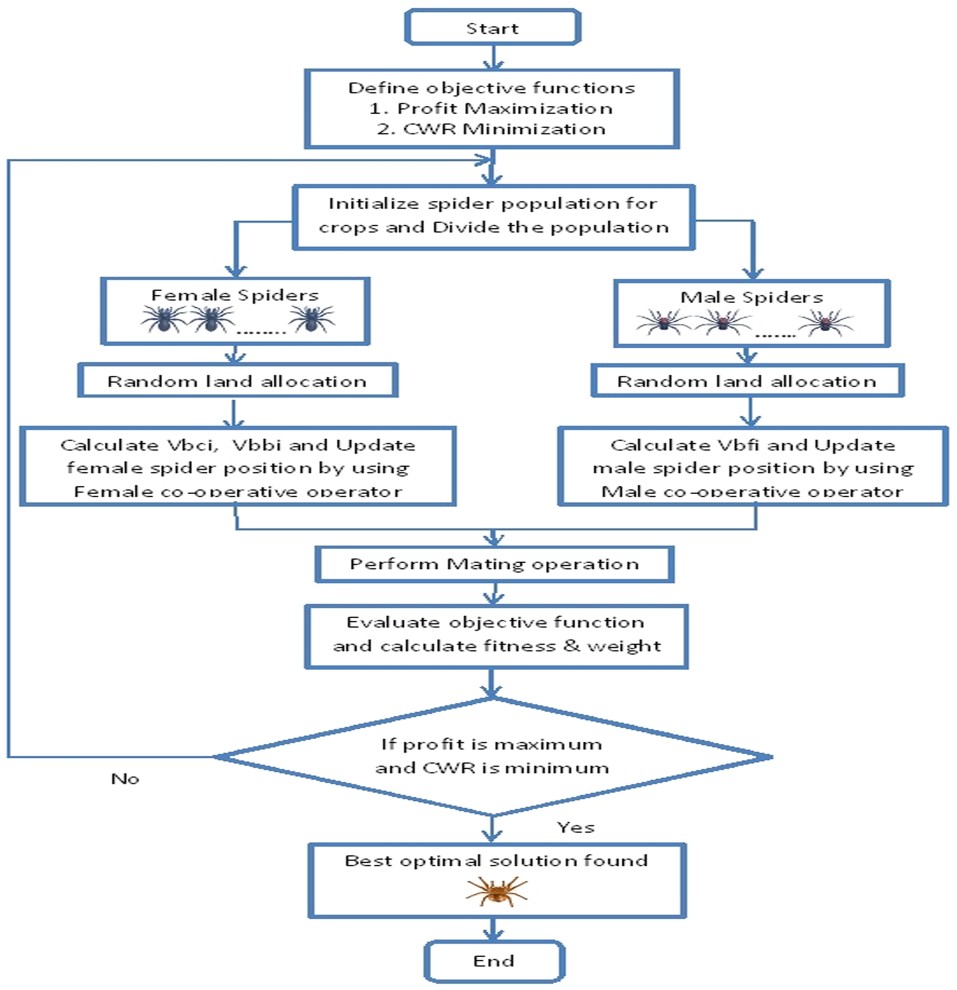

**Figure 2  Flow chart of Social Spider Algorithm for crop planning.**

suggests improvement in total profit and reduction in total water requirement based on the optimal land allocation done by the social spiders. Figure 2 shows the flow chart of SSA for crop planning optimization.

## RESULTS

The implementation of this research work is carried out using C# .Net in Microsoft Visual Studio 2010 and MATLAB R2014b. The results obtained for profit maximization and total water requirement minimization are tabulated in this section. Table 1 list out the crop categories used in this research work. The crops selected are regularly cultivated in Coimbatore region.

Table 2 shows the various types of agricultural land holding based on the area of cultivation. In this research, small land holding of 10,000 m²–20,000 m², small-medium land holding of 20,000 m²–40,000 m², medium land holding of 40,000 m²–1,00,000 m² and large land holding of greater than 1,00,000 m² areas are considered.

**Table 1 Crop categories.**

| S. No | All crops | Cash crops |
|---|---|---|
| 1 | Paddy | Sugarcane |
| 2 | Cholam | Cotton |
| 3 | Maize | Groundnut |
| 4 | Sugarcane | Gingelly |
| 5 | Cotton | Sunflower |
| 6 | Groundnut | – |
| 7 | Gingelly | – |
| 8 | Red gram | – |
| 9 | Black gram | – |
| 10 | Green gram | – |

**Table 2 Classification of agricultural area.**

| Land category | Area of cultivation (in hectare) | Area of cultivation (in m$^2$) |
|---|---|---|
| Marginal | <1 | <10,000 |
| Small | 1–2 | 10,000–20,000 |
| Small-medium | 2–4 | 20,000–40,000 |
| Medium | 4–10 | 40,000–1,00,000 |
| Large | >10 | >1,00,000 |

**Table 3 Test cases used for optimal land allocation.**

| Test case | Land category | Crop category |
|---|---|---|
| Test case 1 | Small | All Crops |
| Test case 2 | | Cash Crops |
| Test case 3 | Small-medium | All Crops |
| Test case 4 | | Cash Crops |
| Test case 5 | Medium | All Crops |
| Test case 6 | | Cash Crops |
| Test case 7 | Large | All Crops |
| Test case 8 | | Cash Crops |

This research is focussed on allocation of optimal land for the two categories of crops, as given in Table 1. Also, the possibilities of attaining maximum profit with minimum water usage are worked out by framing eight test cases, as given in Table 3.

Table 4 shows the optimal land allocation done by SSA in 100 runs for the test cases 1 to 8 with the corresponding profit and water requirements.

Table 5 shows the maximum, minimum, mean, standard deviation and 95% CI (Confidence Interval) of the results obtained by 100 runs of SSA for land allocation, profit and crop water requirement for the eight test cases.

| Table 4 | Optimal crop land allocation by SSA. | | | |
|---|---|---|---|---|
| Test cases | Objective function | Land in m$^2$ | Profit in Rs. | TWR in mm$^3$ |
| Test case 1 | Profit Maximization | 19,850 | 2,033,016 | 348,889 |
| | TWR Minimization | 19,900 | 1,867,616 | 128,472 |
| Test case 2 | Profit Maximization | 19,740 | 2,963,316 | 150,200 |
| | TWR Minimization | 19,640 | 2,708,188 | 104,264 |
| Test case 3 | Profit Maximization | 39,810 | 3,273,473 | 559,598 |
| | TWR Minimization | 39,950 | 2,980,334 | 303,497 |
| Test case 4 | Profit Maximization | 39,870 | 5,222,689 | 335,244 |
| | TWR Minimization | 39,410 | 5,176,876 | 329,079 |
| Test case 5 | Profit Maximization | 99,350 | 8,775,563 | 140,110 |
| | TWR Minimization | 99,650 | 8,358,018 | 126,557 |
| Test case 6 | Profit Maximization | 98,110 | 14,429,256 | 792,379 |
| | TWR Minimization | 99,200 | 12,148,478 | 408,812 |
| Test case 7 | Profit Maximization | 199,930 | 17,161,073 | 2,861,606 |
| | TWR Minimization | 194,580 | 16,858,293 | 2,564,068 |
| Test case 8 | Profit Maximization | 199,200 | 24,272,651 | 1,353,925 |
| | TWR Minimization | 198,890 | 23,814,477 | 1,270,661 |

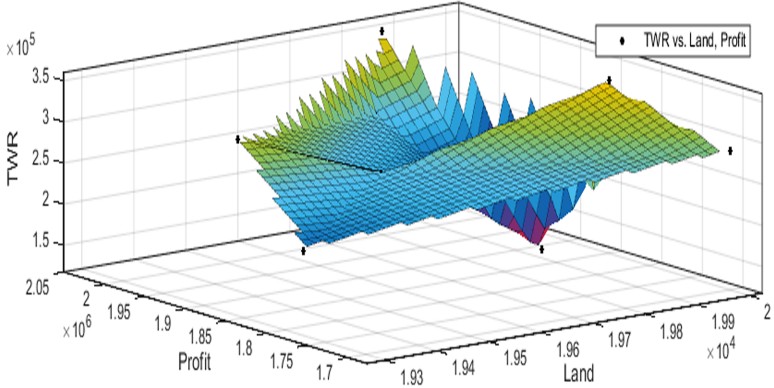

**Figure 3** **Pareto optimal solutions for test case 1.**

Figures 3 to 10 show the performance of SSA in profit maximization and water requirement minimization with respect to various test cases framed.

## DISCUSSION

Profit maximization and water requirement minimization are the two objectives considered in this research work, which are found to contradict each other in most of the results obtained through crop land allocation. Due to the inadequacy of underground water, soil moisture, and less rain fall, there is always a need to supply more water to the agricultural

**Table 5  Maximum, minimum, mean and standard deviation of SSA results.**

| | | Land allocation in m² | Profit in Rs. | TWR in mm³ |
|---|---|---|---|---|
| | Maximum | 19,990 | 2,033,016 | 348,889 |
| | Minimum | 19,300 | 1,693,298 | 128,472 |
| Test case 1 | Mean | 19,738 | 1,861,730 | 266,494 |
| | Std. Dev | 253 | 99,453 | 65,124 |
| | 95% CI | 49 | 19,492 | 12,764 |
| | Maximum | 19,910 | 2,963,316 | 104,264 |
| | Minimum | 19,230 | 2,708,188 | 142,349 |
| Test case 2 | Mean | 19,618 | 2,832,074 | 160,413 |
| | Std. Dev | 223 | 95,162 | 20,569 |
| | 95% CI | 43 | 18,651 | 4,031 |
| | Maximum | 39,950 | 3,273,473 | 589,958 |
| | Minimum | 37,850 | 2,980,334 | 303,497 |
| Test case 3 | Mean | 39,074 | 3,115,553 | 510,561 |
| | Std. Dev | 836 | 98,421 | 81,229 |
| | 95% CI | 163 | 19,290 | 15,920 |
| | Maximum | 39,990 | 5,222,689 | 337,071 |
| | Minimum | 39220 | 5,031,658 | 329,079 |
| Test case 4 | Mean | 39682 | 5,157,520 | 332,686 |
| | Std. Dev | 304 | 58,244 | 3,192 |
| | 95% CI | 59 | 11,415 | 625 |
| | Maximum | 99,960 | 8,775,563 | 1,401,108 |
| | Minimum | 98340 | 8,353,325 | 1,265,573 |
| Test case 5 | Mean | 99373 | 8,554,588 | 1,340,123 |
| | Std. Dev | 566 | 154,113 | 47,349 |
| | 95% CI | 110 | 30,205 | 9,280 |
| | Maximum | 99,910 | 14,429,256 | 884,947 |
| | Minimum | 97,760 | 12,148,478 | 408,812 |
| Test case 6 | Mean | 99,088 | 13,595,435 | 720,347 |
| | Std. Dev | 734 | 768,441 | 138,683 |
| | 95% CI | 143 | 150,611 | 27,181 |
| | Maximum | 199,930 | 17,161,073 | 2,564,068 |
| | Minimum | 194580 | 16,204,427 | 2,891,482 |
| Test case 7 | Mean | 197435 | 16,822,334 | 2,764,595 |
| | Std. Dev | 1863 | 270,128 | 107,563 |
| | 95% CI | 365 | 52,944 | 21,081 |
| | Maximum | 199,980 | 24,272,651 | 1,455,944 |
| | Minimum | 197,840 | 20,619,028 | 1,270,661 |
| Test case 8 | Mean | 198,909 | 2,2610,001 | 1,367,149 |
| | Std. Dev | 773 | 1,180,062 | 59,599 |
| | 95% CI | 151 | 231,287 | 11,681 |
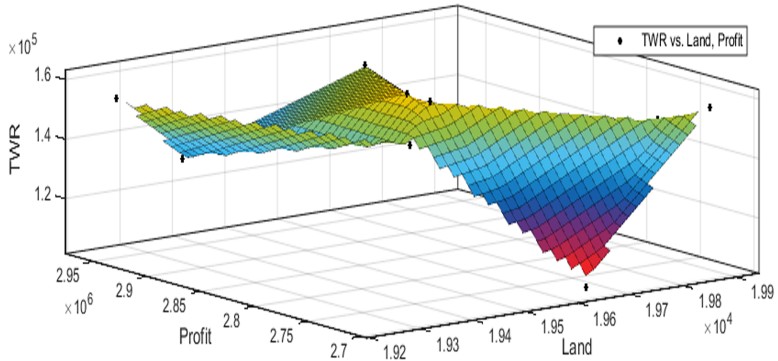

**Figure 4** Pareto optimal solutions for test case 2.

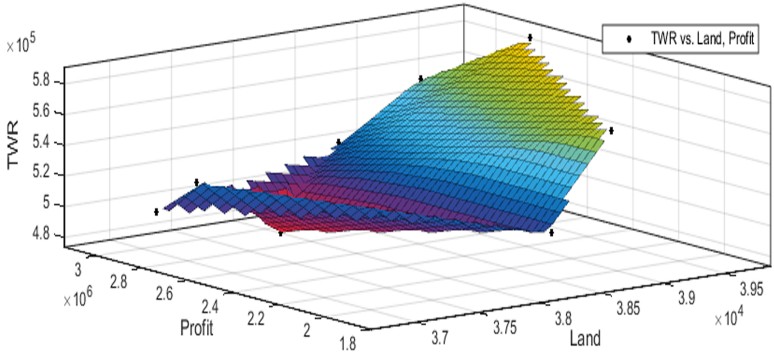

**Figure 5** Pareto optimal solutions for test case 3.

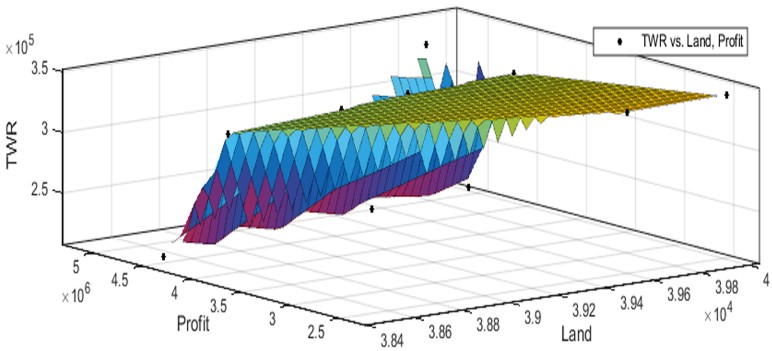

**Figure 6** Pareto optimal solutions for test case 4.

fields to ensure the healthy growth of the crops. This leads to increase in water requirement which poses a heavy challenge to the farmers. Hence this work has paid major attention in water requirement minimization by carefully choosing the crops for cultivation in various land categories in such a way that the farmers are benefited with reasonable profit from

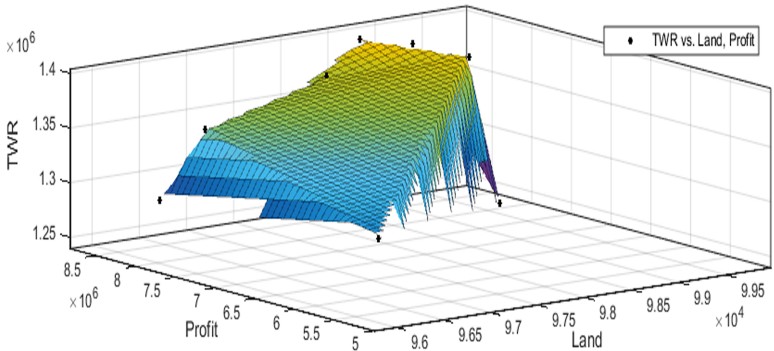

**Figure 7  Pareto optimal solutions for test case 5.**

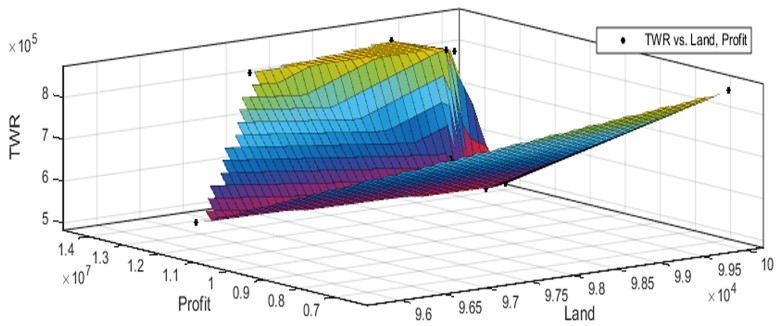

**Figure 8  Pareto optimal solutions for test case 6.**

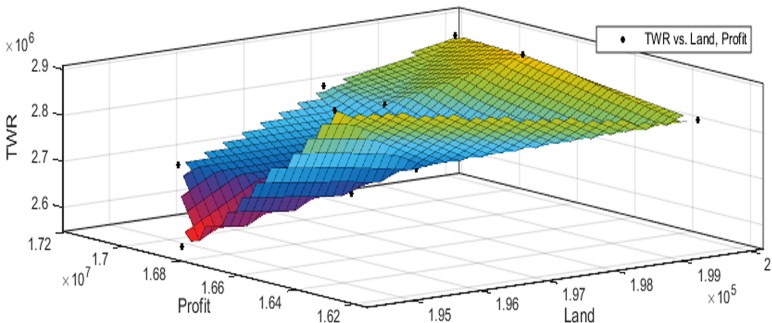

**Figure 9  Pareto optimal solutions for test case 7.**

their crop lands and also be able to manage with the available water resources. Since the problem considered in this research is multi-objective in nature, arriving at Pareto optimal solutions is a challenging task. When the profit obtained from crop land allocation is maximized, water requirement is also found to increase.

Table 4 details upon the optimal land allocation, optimal profit and optimal total water requirement suggested by SSA, that leads to profit maximization as well as water

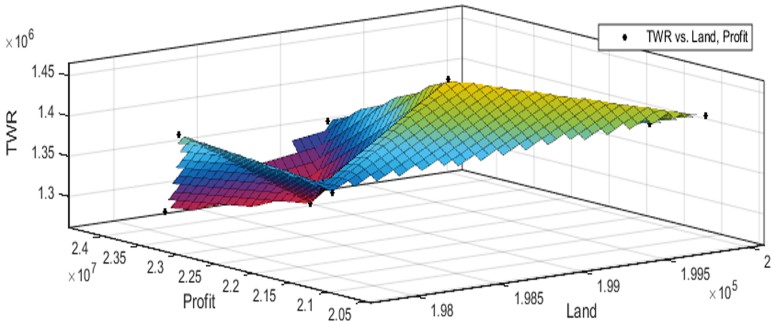

**Figure 10  Pareto optimal solutions for test case 8.**

requirement minimization with respect to each of the eight test cases applied in this work. It could be well observed from these results that maximum profit always require maximum water for cultivation. Similarly, it could also be observed that if the objective of total water requirement minimization is achieved, profit is also minimized. There is always a trade-off in such multi-objective optimization problems. Hence, the SSA algorithm is executed for 100 runs in order to frame a pareto-front and to identify the Pareto optimal solutions, which will satisfy both the objectives to the maximum possible extent without affecting each other (*Antonio et al., 2011*).

These results also highlight the competence of SSA algorithm in producing better profit on an average for all the test cases. Results of test case 1 show that there is 9% of reduction in profit if the water requirement is minimized by 64%. And, the results of test case 2 show that there is 9% of reduction in profit if the water requirement is minimized by 31%. Test cases 1 and 2 deal with small land holdings and for this scenario, optimal land allocation in test case 1 can be recommended, as it achieves good profit with minimal use of water. Results of test case 3 show that there is 9% of reduction in profit if the water requirement is minimized by 45%, and in test case 4, there is 1% of reduction in profit if the water requirement is minimized by 2%. Test cases 3 and 4 deal with small to medium land holdings and for this scenario, optimal land allocation in test case 3 can be recommended, as it achieves better profit with minimal use of water. Results of test case 5 show that there is 5% of reduction in profit if the water requirement is minimized by 10% and in test case 6, there is 15% of reduction in profit if the water requirement is minimized by 50%. Test cases 5 and 6 deals with medium land holdings and for this scenario, optimal land allocation in test case 6 can be recommended, as it achieves good profit with minimal use of water. Results of test case 7 show that there is 2% reduction in profit if the water requirement is minimized by 11%. And, the results of test case 8 show that there is 2% reduction in profit if the water requirement is minimized by 7%. Test cases 7 and 8 deals with large land holdings and in this scenario, optimal land allocation in test case 7 can be suggested where profit is increased without much affecting the water requirement.

The performance of SSA over 100 runs are analysed for each of the test cases and the maximum, minimum, mean, standard deviation and 95% CI values for land allocation, profit and total water requirement are presented in Table 5. The average standard deviation

in optimal profit ranges from minimum 1% to a maximum of 5% and the average standard deviation of optimal water requirement vary from 1% to 20% with respect to all the test cases considered. The 95% CI values state that the results obtained by SSA algorithm can be 95% ascertained. Lesser CI values also indicate the reliability of SSA and the choice of algorithm parameter values. The solutions obtained by SSA are found to be in a consistent search space, which is quite visible by the minimum difference in the standard deviation. The crop planning optimization problem has constantly changing dimensions with respect to the search space, which makes the algorithm to perform challenging exploration and exploitation. It is found that the algorithm has consistently produced optimal results despite the difficulties faced in the identification of feasible solutions. The Pareto optimal solutions obtained by SSA algorithm are visualized from Figs. 3 to 10. The results proved the capability of SSA in obtaining best optimal solutions for crop land allocation in a multi-objective environment.

As stated earlier, the farmland is classified as small, medium, small to medium and large as per the available land area for cultivation. The choice of crops to be grown in the farmland is also suggested by SSA in this work. For small land holdings, all crops category is found to be effective in maximizing the utilization of the available agricultural land, thereby reducing the wastage of land. On the other hand, all crops category grown in small to medium land holdings are effective in maximizing the profit with minimum utilization of water. Cash crops grown in medium land holdings are found to be effective in maximizing the profit with minimum water usage and optimal use of the available agricultural land. Both cash crops and all crops categories grown in large land holdings are found to be equally effective in maximizing the profit with optimum consumption of water and land. It could also be observed that SSA could produce best results in case of small land holdings to medium land holdings, whereas large land holdings pose a challenge in terms of both profit maximization and total water requirement minimization.

## Conclusion and scope for further research

In the present research, an attempt is made to find the optimal land allocation for planting multiple crops in the available land area with water constraints. The optimal land allocation is performed with two objectives, profit maximization and water minimization. Social Spider Algorithm, a new and robust bio-inspired algorithm is applied for optimization and the crops cultivated in the Coimbatore region of Tamilnadu state in India are taken for the case study. Eight test cases are framed based on land availability and crop category. As the land allocation problem is considered as multi-objective problem, the results are analysed for Pareto optimal solutions which dominate both in terms of optimal profit and optimal water requirement for cropping. Rule curves are generated to depict the profit and water requirement with respect to the optimal land allocation for all the test cases analysed in this work. Promising results are obtained which proves the ability of SSA in the optimization of crop planning. Almost 98% of the available land is allocated by the algorithm for cropping, which also shows the effectiveness of the algorithm in minimizing wastage of crop land. Results of this research can be used for cropping recommendations to the farmers with varying land holdings and inadequate water availability. Further optimization techniques in

crop planning such as optimization of soil, labour, transportation, climate, weed, fertilizer and pesticides are planned to be considered for future research.

### Funding
The authors received no funding for this work.

### Competing Interests
The authors declare there are no competing interests.

### Author Contributions
- N Thilagavathi conceived and designed the experiments, performed the experiments, contributed reagents/materials/analysis tools, prepared figures and/or tables, design and Coding.
- T Amudha conceived and designed the experiments, analyzed the data, contributed reagents/materials/analysis tools, prepared figures and/or tables, authored or reviewed drafts of the paper, approved the final draft.

### Data Availability
 The fertilizer optimization code and data is available in the Supplemental Files.

### Supplemental Information
Supplemental information for this article can be found online at http://dx.doi.org/10.7717/peerj.7559#supplemental-information.

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
