# Peer review of "A novel methodology for optimal land allocation for agricultural crops using Social Spider Algorithm"

_PeerJ, doi:10.7717/peerj.7559_

## Round 0.1 · original submission · Major Revisions

In view of the two reviewers' comments, I suggest the author make a major revision based on the reviewers' comments. Specifically, as the first reviewer pointed out, to add a thorough discussion section. The submission will be re-reviewed by the reviewers again, but if the revision is not satisfactory, it will not enter another round of revision.

Reviewer 1 ·

Basic reporting

1. The authors investigated the application of social spider algorithm in the optimal allocation of agricultural land. In general, the contribution is interesting. However, this paper is not well structured and written.
2. At present, this manuscript is like an experimental report, rather than a common research paper.
3. The extensive and methodical experimentation proves the effectiveness of the selected methods. However, the paper would gain in clarity if the achieved results were correlated with the characteristics of the algorithms.
4. In this version, the section of discussion is missing. The author should make the best effort to modify this section. I want to see a considerable improvement in this regard.
5. The introduction section can be revised and structured to improve the flow of ideas from presenting the context, identifying the problem and research gap and presenting the objectives.
6. Need to justify why the social spider algorithm were used specifically.
7. Intensive numeric calculation is involved in this work, but I do not know which platform is employed by the authors. R? Matlab?
8. The research is well defined and relevant although the research gap was vaguely presented (possibly as a result of weak command of the English language).
9. The unit is not included in Tables 4 and 5, which should be modified.
10. The map in the Figure 1 is not complete, some basic elements (north arrow, scale bar and other legends) are missing.
11. The figures are neither complete nor clear, some details of theses curves are lost.

Experimental design

no

Validity of the findings

no

·

Basic reporting

1. I suggest to rephrase the title as "A Novel Methodology for Optimal Land Allocation for Agricultural Crops using Social Spider Algorithm"
2. Line 32: Suggest rephrasing the word economy as revenue.
3. Equation 2: Inconsistency in using variable. Equation 2 contains a variable, "Productivityi", whereas in description (Line No: 106) uses "proi".
4. Maintain consistency in claiming the scope of the variable. For example, in Line No: 111 says seed cost whereas Line No: 114 says expense on seeds. I suggest use of variable names used in Equation 3 within brackets while describing them in Line Nos: 111 & 112 (if applicable).
5. Double check the citation format in Line No: 144.

Experimental design

Even though the authors have explained very well and established the social spider algorithm (SSA) to solve the problem, a couple of sentences required to be included to relate the chosen algorithm (SSA) to the problem applied.

Validity of the findings

No comment

Additional comments

Use of Figure 1: Does it have any relevance to the research? I suggest you to remove Figure 1 if its of no relevance.

---

## Round 0.2 · accepted · Accept

The authors have apparently addressed the reviewers' concerns. To that end, I can now recommend acceptance of the current manuscript.

Reviewer 1 ·

Basic reporting

As I can see from the previous review, you have corrected all of the reviwers comments. It is ready for publication.

Experimental design

-

Validity of the findings

-

Additional comments

-

·

Basic reporting

No comment

Experimental design

No comment

Validity of the findings

No comment